# Model Comparisons of Flow and Chemical Kinetic Mechanisms for Methane–Air Combustion for Engineering Applications

**Di He** 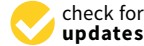, **Yusong Yu \*, Yucheng Kuang and Chaojun Wang \***

Institute of Combustion and Thermal Systems, School of Mechanical, Electronic and Control Engineering, Beijing Jiaotong University, Beijing 100044, China; 19116007@bjtu.edu.cn (D.H.); 18116012@bjtu.edu.cn (Y.K.)
\* Correspondence: ysyu@bjtu.edu.cn (Y.Y.); cjwang@bjtu.edu.cn (C.W.)

**Abstract:** The reasonably accurate numerical simulation of methane–air combustion is important for engineering purposes. In the present work, the validations of sub-models were carried out on a laboratory-scale turbulent jet flame, Sandia Flame D, in comparison with experimental data. The eddy dissipation concept (EDC), which assumes that the molecular mixing and subsequent combustion occur in the fine structures, was used for the turbulence–chemistry interaction. The standard $k$-$\varepsilon$ model (SKE) with the standard or the changed model constant $C_{1\varepsilon}$, the realizable $k$-$\varepsilon$ model (RKE), the shear-stress transport k-$\omega$ model (SST), and the Reynolds stress model (RSM) were compared with the detailed chemical kinetic mechanism of GRI-Mech 3.0. Different reaction treatments for the methane–air combustion were also validated with the available experimental data from the literature. In general, there were good agreements between predictions and measurements, which gave a good indication of the adequacy and accuracy of the method and its further applications for industry-scale turbulent combustion simulations. The differences between predictions and measured data might have come from the simplifications of the boundary settings, the turbulence model, the turbulence–reaction interaction, and the radiation heat transfer model. For engineering predictions of methane–air combustion, the mixture fraction probability density function (PDF) model for the partially premixed combustion with RSM is recommended due to its relatively low simulation expenses, acceptable accuracy predictions, and quantitatively good agreement with the experiments.

**Keywords:** eddy dissipation concept (EDC); Sandia Flame D; methane–air combustion; chemical kinetic mechanism; probability density function (PDF)

## 1. Introduction

The consumption of natural gas can produce 50% less pollution than other fossil fuels [1], and it accounts for 23.7% of global energy consumption [2]. It has also been reported that natural gas will gradually take the place of coal as an important fuel for power generation [3], and many natural gas burners have been designed and modeled [4,5]. In these applications of industry-scale utilizations, predictive tools like computational fluid dynamics (CFD) represent an effective and economical approach for the burner design and optimization. However, in terms of the turbulent flow, gas-phase chemical reactions, heat transfer, etc., the accurate numerical simulation of natural gas combustion is usually competitive. Many turbulent models, including the standard $k$-$\varepsilon$ model (SKE) [6], the realizable $k$-$\varepsilon$ model (RKE) [7], the shear-stress transport (SST) $k$-$\omega$ model [8], and the Reynolds stress model (RSM) [9–11]—even the detached eddy simulation (DES) model [12,13] and the large eddy simulation (LES) model—embedded in commercial computational fluid dynamics CFD software, can address the turbulence flow well [14,15].

For turbulence–chemistry interaction modeling in computational fluid dynamics (CFD), many models, including the eddy break up (EBU) [16], the eddy dissipation concept (EDC) [17,18], the mixture fraction probability density functions (PDF) [19–22], the conditional moment closure (CMC) models [23,24], and various types of flamelet models [25–27], have been developed.

An extended version of the EBU approach, known as the EDC, has been developed to incorporate detailed chemical kinetics in turbulent flows. This model has shown adequate predictions for premixed, partially premixed, and non-premixed combustion regimes [26]. As one of the most commonly adopted approaches in the modeling of turbulent reacting flows in the context of the steady/unsteady compressible Reynolds-averaged Navier–Stokes equations (RANS/URANS), which sometimes referred to as Reynolds-averaged simulations (RAS), the EDC was initially developed in the 1970s [17,18]. The EDC was then formulated as a well-established turbulent combustion closure model in the 1990s−2000s [28–30]. The EDC has been successfully applied to the numerical simulation of combustion for a long period of time [30,31], and it is implemented in most available commercial CFD codes. Various models [15,32–39] have been validated with the Flame D setup [40–43] with the detailed chemical kinetic mechanism of GRI-Mech 3.0 [44] (consisting of 53 species and 325 elemental reactions, abbreviated as 53–325) or the reduced mechanisms based of GRI-Mechs 1.2 [45] and 2.11 [46] under steady-state species assumptions. For example, sp21 and sp24 [47,48] are from GRI-Mech 1.2, and two augmented reduction mechanisms, ARM 9 [49] and ARM 19 [49], are reductions of the GRI-Mech 2.11. In addition to the work of GRI-Mechs, other mechanisms have been also developed for methane–air combustion simulations, such as the skeletal mechanism [50,51] (6–41), KH97 [52] (28–104), COMB [53] (49–236), K97 [54] (57–353), and SKG04 [55] (72–520). Methane–air combustion was once described by a four-step global reaction mechanism with six species and four reactions [32,56] for engineering applications. Though this global reaction mechanism was proven to strongly predict CO concentrations, it fails in predicting $H_2$ and $H_2O$, especially in fuel-rich conditions [32,54].

For applications of industry-scale natural gas combustion furnaces, the computational domains are, however, often much bigger (containing more grid cells) than those in experimental burners used to validate combustion models. This makes simulations with detailed chemistry and complicated turbulence models, such as GRI-Mech 3.0, DES, and LES, very time consuming and sometimes impossible; therefore, a computationally inexpensive treatment of the chemical kinetics and turbulence models is sought for engineering applications. This work intends to provide a validated basis for these applications with acceptable expenses.

## 2. Object Description

The Flame D from the Sandia/TNF workshop, as shown schematically in Figure 1, is a piloted methane–air diffusion flame [40–42]. The central main jet consists of a methane–air mixture (with 25% by volume of $CH_4$) corresponding to an equivalence ratio of 3.174. This is above the upper flammability limit of methane so combustion is still controlled by mixing. It was surrounded by a pilot flame and a slow co-flow of air outside. Flame D exhibits local extinction to a limited degree [37,40]. Here, the pilot flame was burning a mixture of $C_2H_2$, $H_2$, air, $CO_2$, and $N_2$ with an enthalpy and equilibrium composition that was equivalent to a mixture of methane and air at an equivalence ratio of $\varphi = 0.77$. The experimental data were documented in detail by Barlow et al. [42].

This flame was predicted by means of 2D steady RANS. The settings of the solver in all the simulations included a SIMPLE scheme for the pressure–velocity coupling, Green–Gauss based for the gradient, Presto! for the pressure, and second order upwind for the others. The thermodynamic models used in this work for EDCs were from GRI Mech, and the model for PDF was from the Fluent package. The computational domain extended from $5d$ behind the nozzle exit plane to $100d$ in the axial direction and $50d$ in the radial direction, where $d$ stands for the main jet diameter. The 2D RANS simulations were carried out on a structured grid with an increased resolution close to the nozzle and the symmetry axis. The jet velocity at the nozzle exit was approximated by including the nozzle with length of $5d$ in the computational domain and axial velocity setting at the inflow boundary using the velocity profiles measured at the nozzle exit plane. The CFD simulations of Flame D were conducted on an axisymmetric numerical mesh. Though not shown in Figure 1,

the following four meshes were used for comparison: A: 49 × 50; B: 157 × 71; C: 202 × 88; and D: 252 × 118. All the meshes gave very close results, and either could be used to produce a grid-independent solution. However, the finer mesh (mesh C, 202 × 88 shown in Figure 1) with finer grids in the areas of the jet and the pilot was selected and used in all the simulations even though other grids could be used for this 2D steady work. Considering the model selections were for the engineering methane–air combustion, a much finer mesh was unnecessary.

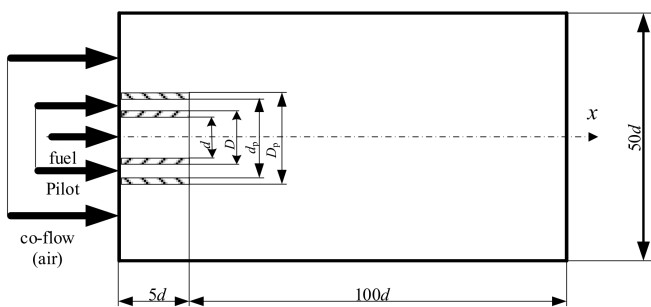

(**a**) Schematic configuration of Sandia Flame D.

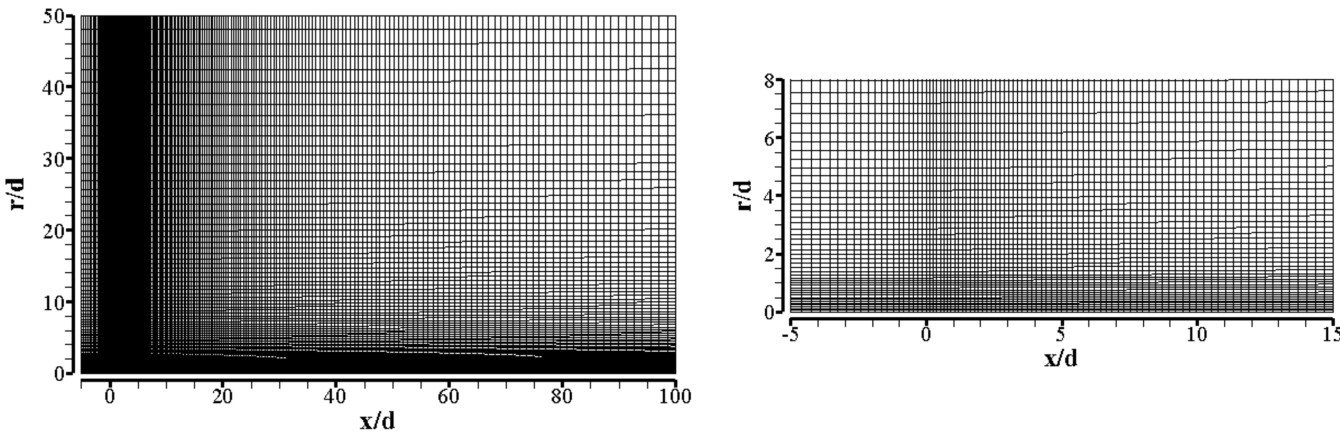

(**b**) The grid for the computational domain.  (**c**) The details of the grid near the inlet.

**Figure 1.** Schematic configuration of the Sandia Flame D and the grids.

The burner geometry data and boundary conditions used for the simulations are shown in Table 1. It should be noted that the jet velocity, co-flow velocity, pilot velocity, and temperature in the experiment were in a range [42]. For example, the jet bulk velocity, the pilot bulk velocity, and the air co-flow velocity in the experiment were 49.6 ± 2, 11.4 ± 0.5, and 0.6 ± 0.05 m/s, respectively, and these data for the velocity inputs were averaged as 49.7, 12.3, and 0.8 m/s, respectively, from the profiles. The pilot temperature in the experiment was 1880 ± 50 K, while the setting was 1880 K. The profiles for the jet, pilot, and co-flow with the average values, shown in Table 1, were used in the simulation, and these settings, of course, induced different predictions with the experiments. The profiles for the boundary conditions of the average velocity and its fluctuations [42,43], as well as the turbulent kinetic energy [43], are shown in Figure 2. The turbulent dissipation rate $\varepsilon$ and the specific dissipation rate $\omega$ used for the turbulence models could be estimated from the average and the fluctuations.

**Table 1.** Burner geometry data and boundary conditions for the Sandia Flame D.

| Item | Unit | Values in Experiment | Setting Values |
|---|---|---|---|
| Jet mixture $CH_4$/air | vol% | | 25/75 |
| Pilot mixture composition, mass fraction | % | | See [42] |
| $N_2$ | | | 73.42 |
| $O_2$ | | | 5.40 |
| O | | | $7.47 \times 10^{-2}$ |
| $H_2$ | | | $1.29 \times 10^{-2}$ |
| H | | | $2.48 \times 10^{-3}$ |
| $H_2O$ | | | 9.42 |
| CO | | | 0.407 |
| $CO_2$ | | | 10.98 |
| OH | | | 0.28 |
| NO | | | $4.8 \times 10^{-4}$ |
| Main jet inner diameter, $d$ | mm | | 7.2 |
| Pilot annulus inner diameter, $D$ | mm | | 7.7 |
| Pilot annulus outer diameter, $d_p$ | mm | | 18.2 |
| Burner outer diameter, $D_p$ | mm | | 18.9 |
| Jet bulk velocity | m/s | $49.6 \pm 2$ | Profile; see [42] |
| Jet inlet turbulent kinetic energy | $m^2/s^2$ | | Profile; see [43] |
| Jet inlet turbulent dissipation rate | $m^2/s^3$ | - | Profile; estimated with Equation (1) |
| Jet inlet specific dissipation rate | 1/s | | Profile; estimated with Equation (2) |
| Pilot bulk velocity | m/s | $11.4 \pm 0.5$ | Profile; see [42] |
| Pilot inlet turbulent kinetic energy | $m^2/s^2$ | | Profile; see [43] |
| Pilot inlet turbulent dissipation rate | $m^2/s^3$ | - | Profile; estimated with Equation (1) |
| Pilot inlet specific dissipation rate | 1/s | | Profile; estimated with Equation (2) |
| Air co-flow velocity | m/s | $0.9 \pm 0.05$ | Profile; see [42] |
| Air co-flow inlet turbulent kinetic energy | $m^2/s^2$ | | Profile; see [43] |
| Air co-flow inlet turbulent dissipation rate | $m^2/s^3$ | - | Profile; estimated with Equation (1) |
| Air co-flow inlet specific dissipation rate | 1/s | | Profile; estimated with Equation (2) |
| Jet temperature | K | | 294 |
| Pilot temperature | K | $1880 \pm 50$ | 1880 |
| Co-flow temperature | K | | 291 |
| Reynolds number, $Re_{jet}$ | - | | 22,400 |

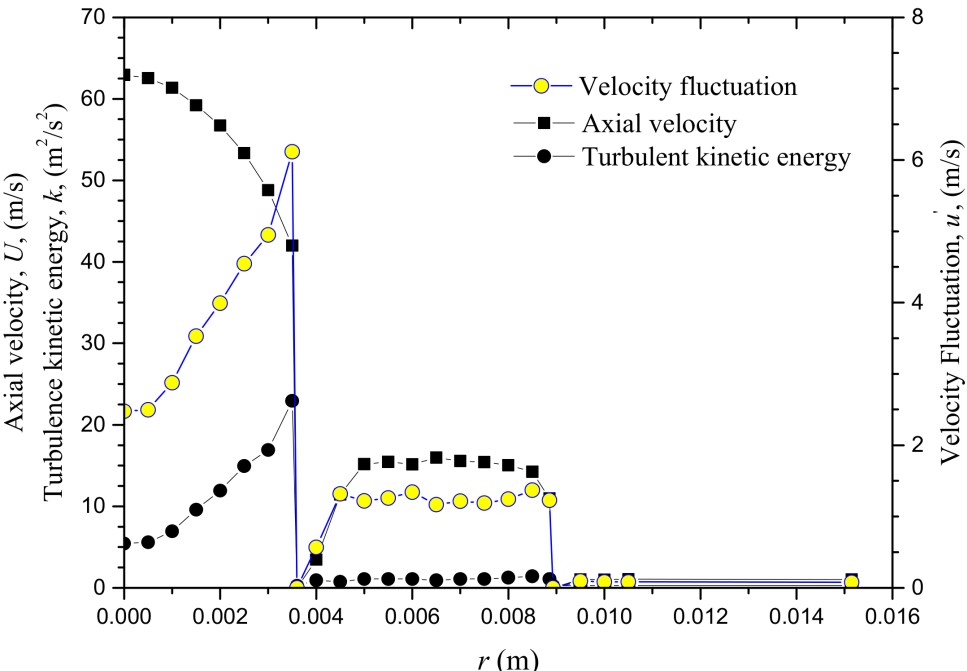

**Figure 2.** Inlet boundaries for the Sandia Flame D.

The turbulent dissipation rate, $\varepsilon$, at the inlets can be estimated as:

$$\varepsilon = \frac{k^{3/2}}{\ell} \tag{1}$$

where $k$ is turbulent kinetic energy and $\ell$ is the turbulence length scale.

The specific dissipation rate, $\omega$, used for the turbulence models, can be estimated as:

$$\omega = \frac{k^{1/2}}{C_\mu \ell} \tag{2}$$

where the value of $C_\mu$ is 0.09.

## 3. Predictions with Different Turbulence Models

The validation procedure concerned the modelling of turbulence, for engineering purposes, with a detailed chemical kinetic mechanism. In order to support the choice of the associated turbulence models, simulations of the laboratory non-premixed turbulent flame, the Sandia Flame D [40–42], were performed. For the turbulence modelling, the Reynolds averaged Navier–Stokes approach (RANS) was applied to the simulations of the Sandia Flame D to estimate the influence of the accuracy of the turbulence modelling on the results. SKE, RKE, SST, and RSM turbulence models were compared in the first step because they can be used in engineering applications with acceptable expenses.

For turbulence–chemistry interaction modeling, the EDC with the detailed chemical kinetic mechanism of GRI-Mech 3.0 was applied to the RANS simulations. The general requirements set on the CFD model predetermined the selection of the EDC to a certain degree. The investigations carried out served for the evaluations of the EDC and the RANS models considering the simulations of gas phase combustion.

All simulations were carried out with the commercially available software ANSYS Fluent 19 [57]. To speed-up the CPU-intensive treatment of the detailed reaction kinetics, the EDC was applied in conjunction with the in-situ adaptive tabulation (ISAT) algorithm [58]. The method is based on a run-time tabulation of the local (realized) region of the chemical state space (species concentrations, temperature, and pressure) accessed during the CFD

simulation. ISAT combines both direct integration for new entries and linear interpolation between already directly integrated values within a specified interpolation error tolerance. The interpolation error tolerance determines the size of the ISAT table and the time needed for its generation. In addition, it strongly influences the accuracy of simulation results.

It is important to choose adequate error tolerances for an ISAT table and for integrating the ordinary differential equations (ODEs) of chemical rates. The experiences of Masri et al. [59] suggested that the ISAT and the ODE error tolerances should be not greater than $6.25 \times 10^{-6}$ and $1 \times 10^{-8}$, respectively. Final error tolerances of $6.25 \times 10^{-6}$ and $1 \times 10^{-12}$ were therefore used in all the calculations reported below. The default settings were used for the EDC constants.

Turbulence modeling is known to have considerable effects on combustion results. Sensitivity to turbulence models was investigated by applying SKE, RKE, or RSM with an EDC-based detailed chemical kinetic mechanism and comparing it with experimental data [40–42] to determine the candidate turbulence model to be used in the simulations of industry-scale natural gas combustion furnaces. The standard constant values of SKE, $C_{1\varepsilon}$ and $C_{2\varepsilon}$, result in an over-prediction of spreading and turbulence diffusion of round jet flows [60]. This tends to increase downstream from a nozzle. The over-predicted mixing also exaggerates fuel conversion and hence affects the location of heat release and the development of density and the velocity fields. To reduce spreading/diffusion, various modifications have been suggested. The main approach is to change $C_{1\varepsilon}$ or $C_{2\varepsilon}$ [31,59–63]. Hence, for combustion-model investigations, although not in general, it was appropriate to make "ad hoc" modifications of the turbulence models to give the best representation of the turbulent flow field. Here for the SKE model, the constant $C_{1\varepsilon}$ of the $\varepsilon$-equation was changed from 1.44 (the standard value) to 1.6, and this model is referred as SKE-R hereafter. This change is a standard correction proposed for this model to improve prediction of the jet spreading rate [34,59].

The turbulence models, i.e., SKE, SKE-R, RKE, SST, and RSM, were validated with the EDC-based detailed chemical kinetic mechanism, identified in the results as EDC-53/SKE, EDC-53/SKE-R, EDC-53/RKE, EDC-53/SST, and EDC-53/RSM, respectively. Following standard best practice, radiation heat transfer was approximately solved using discrete ordinate (DO) radiation equation with 100 ($4 \times 5 \times 5 \times 1$) directions along with the weighted sum of gray gas model (WSGGM) [64] for absorption coefficient. For a more accurate solution of a radiation equation, an efficient exponential wide band (E-EWB) model [65–67] with a user-defined function (UDF) or full-spectrum *k*-distribution (FSK) look-up table and its improvement [68–73] can be used. The emphasis of this work is on the choices of the turbulence models and the chemical kinetic mechanisms for engineering modeling purposes.

The main settings of the turbulence models are shown in Table 2. The comparisons were done with the only difference of the turbulence models used for the five cases. The EDC was used to model the turbulence–chemistry interaction with the default EDC constants. A comparison of the predicted flame temperatures is displayed in Figure 3. The predicted flame construction by the turbulence models of SKE, RKE, and SST were found to be quite similar. The flame shape predicted by SKE-R (shown in Figure 3b) was the longest and apparently different from the other four, and the results might not have been correctly predicted by the SKE-R model. This implied that the chemical reactions are strongly affected by the turbulent flows. The flame shape predicted by RSM, shown in Figure 3e, was shorter than that by SKE-R but longer that those by others. Scalar data are presented in Figures 4 and 5. Mean temperature, turbulent kinetic energy, and main mean composition profiles along the axis are shown in Figure 4 for the Sandia Flame D with the turbulence models of SKE, SKE-R, RKE, SST, and RSM. Mean axial velocity profiles are shown in Figure 5 for the Sandia Flame D with the turbulence models.

**Table 2.** Main settings of turbulence models.

| Case | Turbulence Models | Options | Near-Wall Treatment | Model Constants |
|---|---|---|---|---|
| EDC-53/SKE | Standard $k$-$\varepsilon$ model | Viscous heating; production limiter | | |
| EDC-53/SKE-R | Standard $k$-$\varepsilon$ model with changed constant $C_{1\varepsilon}$ | Viscous heating; production limiter | | |
| EDC-53/RKE | Realizable $k$-$\varepsilon$ model | Viscous heating; production limiter | Standard wall functions | Defaults |
| EDC-53/SST | Shear-stress transport (sst) $k$-$\omega$ model | Low-Re correction; viscous heating; production limiter | | |
| EDC-53/RSM | Reynolds stress model | Quadratic pressure-strain; Wall BC from $k$ equation | | |

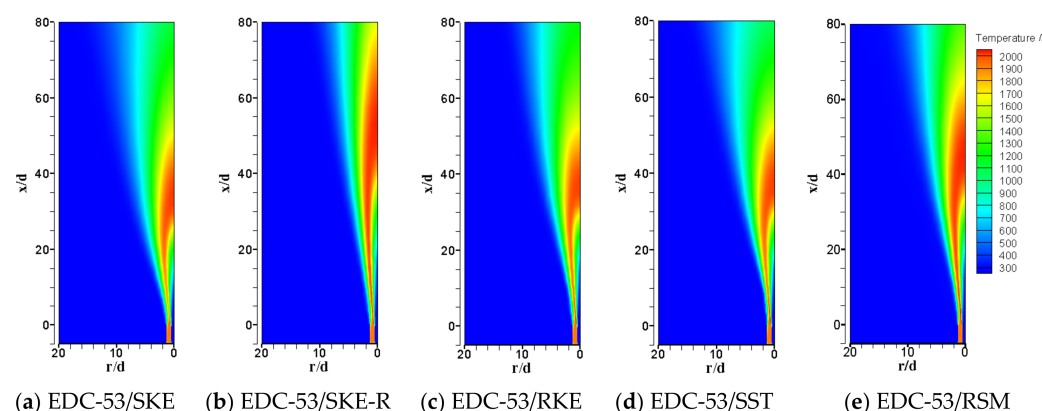

(**a**) EDC-53/SKE  (**b**) EDC-53/SKE-R  (**c**) EDC-53/RKE  (**d**) EDC-53/SST  (**e**) EDC-53/RSM

**Figure 3.** Iso-contours of mean temperature with turbulence models of SKE, SKE-R, RKE, SST, and RSM for the Sandia Flame D with the detailed chemical kinetic mechanism of GRI-Mech 3.0.

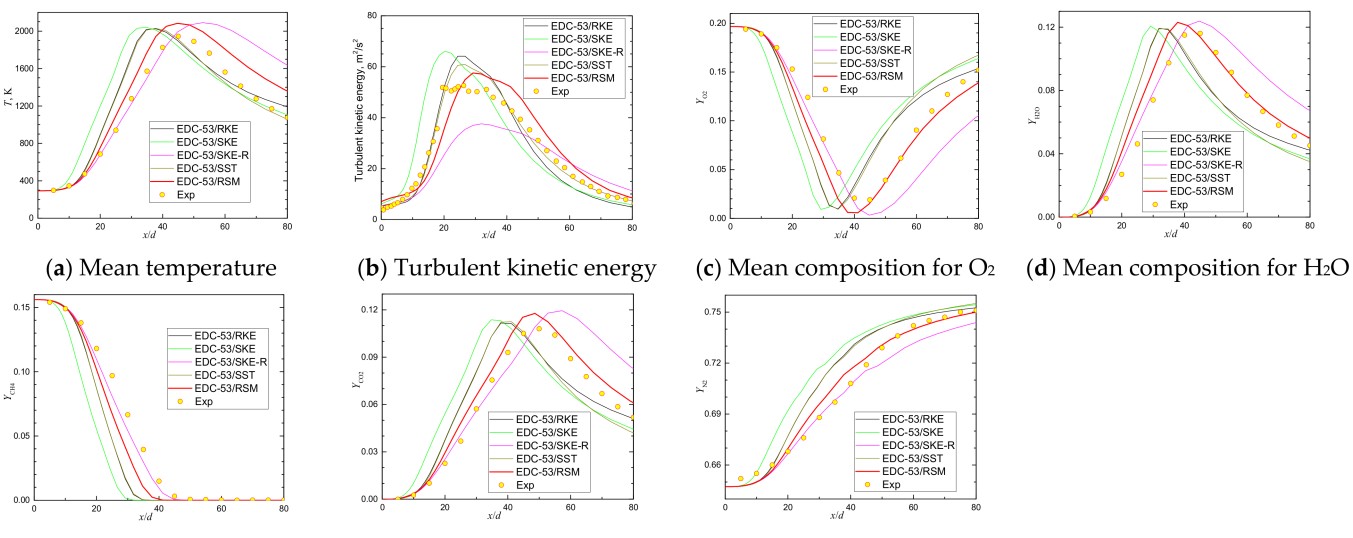

(**a**) Mean temperature  (**b**) Turbulent kinetic energy  (**c**) Mean composition for O$_2$  (**d**) Mean composition for H$_2$O

(**e**) Mean composition for CH$_4$  (**f**) Mean composition for CO$_2$  (**g**) Mean composition for N$_2$

**Figure 4.** Comparison of the turbulence models of SKE, SKE-R, RKE, SST, and RSM for the Sandia Flame D: mean temperature, turbulent kinetic energy, and main mean composition profiles along the axis with the detailed chemical kinetic mechanism of GRI-Mech 3.0.

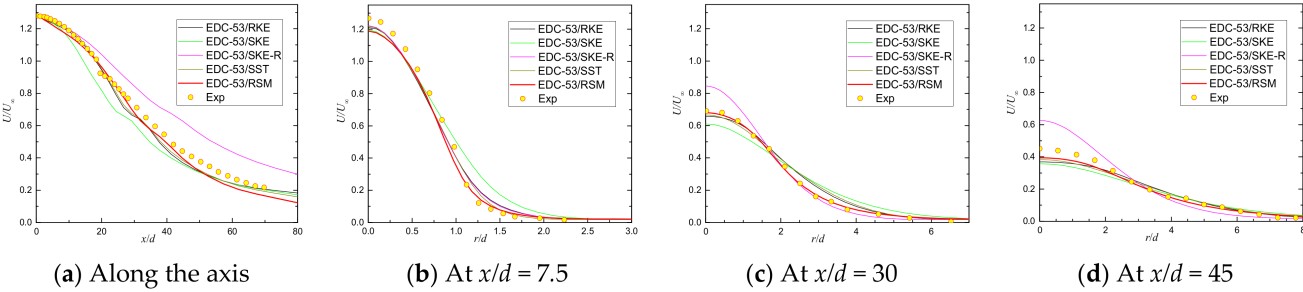

**Figure 5.** Comparison of the turbulence models of SKE-R, SKE, RKE, SST, and RSM for the Sandia Flame D: mean axial velocity profiles with the detailed chemical kinetic mechanism of GRI-Mech 3.0.

Since the used chemistry and the turbulence–chemistry interaction model were the same, the differences shown in Figures 3–5 must have mainly resulted from the turbulent flow models. It is well-known that round-jet results for RANS, in particular for two-equation models but also for multi-equation models, depend on the choice of model constants. The present results indicated that the challenges were very related to turbulent flow modelling. The results shown in Figures 3–5 indicate that the standard adjustment of increasing one of the constants, $C_{1\varepsilon}$, from 1.44 (used in the SKE model) to 1.6 (used in the SKE-R model) was not effective in this work or the work by Masri et al. [59]. This was due to the lower density of the co-flow. Differences in density between the jet, pilot, and co-flow are known to affect the mixing behavior of jets.

A reasonable qualitative prediction of the flame was achieved for all model combinations. However, in contrast to those results from LES simulations [15,33–35,37,38], none of them came quantitatively close to the measured data [40–43], shown with the yellow-filled red circles in Figures 4 and 5. This means the predictions with the turbulence models used in this work could not exactly reproduce the methane–air combustion in the Sandia Flame D. Deviations of predicted and measured values are shown in Table 3 for different turbulence models with GRI-Mech 3.0, where it can be seen that the largest deviation happened to the minor species, i.e., OH.

**Table 3.** Deviations of predicted and measured values for different turbulence models with GRI-Mech 3.0 (deviation = (prediction − measurement)/measurement × 100%).

| Scalar Peak and Its Location | Exp | EDC-53/SKE | Deviation | EDC-53/SKE-R | Deviation | EDC-53/RKE | Deviation | EDC-53/SST | Deviation | EDC-53/RSM | Deviation |
|---|---|---|---|---|---|---|---|---|---|---|---|
| Max Temperature@axis | 1945 | 2040 | 4.88% | 2090 | 7.47% | 2026 | 4.16% | 2031 | 4.41% | 2083 | 7.08% |
| Location@$x/d$ = | 45.00 | 34.74 | | 52.79 | | 37.83 | | 37.83 | | 44.75 | |
| Max Turbulent energy@axis | 52.57 | 66.07 | 25.68% | 37.58 | −28.52% | 64.12 | 21.98% | 60.87 | 15.80% | 57.50 | 9.38% |
| Location@axis, $x/d$ = | 26.28 | 20.30 | | 31.86 | | 24.42 | | 26.72 | | 29.20 | |
| Min YO$_2$@axis | 0.019 | 0.009 | −51.51% | 0.003 | −82.82% | 0.010 | −48.76% | 0.009 | −51.56% | 0.006 | −69.93% |
| Location@axis, $x/d$ = | 45.00 | 29.20 | | 44.75 | | 34.74 | | 34.74 | | 41.16 | |
| Max YH$_2$O@axis | 0.116 | 0.121 | 3.96% | 0.124 | 6.71% | 0.119 | 2.84% | 0.119 | 2.57% | 0.123 | 6.02% |
| Location@axis, $x/d$ = | 45.00 | 29.20 | | 44.75 | | 31.86 | | 34.74 | | 37.83 | |
| Max YCO$_2$@axis | 0.108 | 0.114 | 5.27% | 0.119 | 10.59% | 0.112 | 3.42% | 0.113 | 4.17% | 0.118 | 8.95% |
| Location@axis, $x/d$ = | 50.000 | 34.74 | | 57.27 | | 37.83 | | 41.16 | | 48.62 | |
| Max YCO@axis | 0.0453 | 0.0514 | 13.57% | 0.0588 | 29.88% | 0.0529 | 16.73% | 0.0523 | 15.39% | 0.0565 | 24.62% |
| Location@axis, $x/d$ = | 40.00 | 26.72 | | 41.16 | | 31.86 | | 31.86 | | 37.83 | |
| Max YH$_2$@axis | 0.00288 | 0.00290 | 0.69% | 0.00346 | 20.21% | 0.00287 | −0.35% | 0.00285 | −0.96% | 0.00343 | 19.07% |
| Location@axis, $x/d$ = | 40.00 | 26.72 | | 41.16 | | 29.20 | | 29.20 | | 34.74 | |
| Max YOH@axis | 0.00148 | 0.00441 | 197.74% | 0.00389 | 162.92% | 0.00433 | 192.36% | 0.00426 | 187.95% | 0.00401 | 171.18% |
| Location@axis, $x/d$ = | 50.00 | 34.74 | | 52.79 | | 37.83 | | 37.83 | | 44.75 | |

Taking the temperature peak of experiment as the demarcation point in Figure 4a, before this point, the temperature profile predicted by SKE-R was consistent well with the range of experiment data. However, after this point, the temperature was largely over-predicted. Only the temperature profile predicted by RSM reasonably agreed with the experiment data, and the peak position was properly predicted at $x/d$~45. All the temperature peaks were slightly over-predicted, as shown in Table 3, and the largest was about 145 K by SKE-R. The temperature peak locations predicted with SKE (~2040 K@$x/d$~35), RKE (~2026 K@$x/d$~38), and RSM (~2055 K@$x/d$~38) were shifted towards the burner (Figure 4a), and the peak with SKE-R (~2090 K@$x/d$~53) was shifted downwards from the burner a little (Figure 4a) compared to the experiments (~1945 K@$x/d$~45). These deviations and others, shown in Figures 4 and 5 and Table 3, have also been found by other predictions [34,37]. The temperature peak location predicted with RSM (~2083 K@$x/d$~45) agreed well with the experiment location, with an over-prediction of approximately 138 K. Compared with the results by RSM and SKE-R, the tuning of the SKE model constant yielded a worse peak location, a better prediction of the temperature sloped up to its peak, and large deviations between the predictions and the experiments appeared downwards of the peak. However, an increase of the production term in the $\varepsilon$-equation by tuning the standard model constant from 1.44 (SKE) to 1.6 (SKE-R) led to a considerable decrease of the turbulent kinetic energy, shown in Figure 4b, and thus to a very poor agreement between the predicted and the experimental results in this work. Therefore, predictions by SKE-R could be ignored. Compared to the experiments of the turbulent kinetic energy, the best results were reached by the RSM predictions. It should also be noted that the choice of turbulence model affected the prediction of the peak values and the peak locations of the scalars, as shown in Table 3. For the predictions of major species, the largest deviation was found for the peak values of the $O_2$ mass fraction. Checking the minimum $O_2$ mass fraction profiles shown in Figure 4c demonstrates that they are acceptable, especially that by RSM. The differences between simulations by RSM and measured data were not significant for the temperature and the main species such as $O_2$, $H_2O$, and $CO_2$. The behavior of these species had the same trend as that for the temperature. The profiles of the values for composition along the axis replicated the behavior of the temperature. The peak of the mean mass fraction of $CO_2$ (Figure 4f) was largely over-predicted by SKE-R compared to those by the other four models. The mean mass fraction of $CH_4$ along the axis, predicted by SKE-R shown in Figure 4e, occasionally agreed quite well with the experiments. However, those predicted by the others shown in Figure 4e were under-predicted, especially close to the temperature peak, implying that $CH_4$ along the axis was burnt out earlier than in the experiments due to the strong mixing and spreading. If the results from SKE-R were ignored, the $CH_4$ predictions by RSM agreed best with the experiments among the predictions. The mean mass fraction of $N_2$ along the axis, predicted by SKE-R shown in Figure 4g, was under-predicted compared to those predicted by the other four turbulence models. The best predictions were done with RSM, while about half of the predicted $N_2$ results by SKE-R were again very close to the experiments.

With the exceptions of SKE-R and SKE, differences between the simulations and the measured data were not significant for the mean axial velocities, as shown in Figure 5. Very small differences were found from the predicted results by the turbulence models of RKE, SST, and RSM—identified as EDC-53/RKE, EDC-53/SST, and EDC-53/RSM, respectively, in Figure 5—with the EDC-based detailed chemical kinetic mechanism. All the predictions by RKE and SST agreed very well with each other. The predicted scalar data by RSM and the experiments, shown in Figures 4 and 5, were very close. Therefore, the RSM model was selected for the turbulence simulation of the Sandia Flame D in all further calculations for its relatively better predictions.

The disagreement between the measured and the computed temperatures on the centerline, with the exceptions of the inconsistency of the boundary conditions and the turbulence modeling (particularly near the jet exit plane), was largely due to the fact that the heat transfer from the hot co-flow was rather significant and yet not properly accounted

for in the computations. The mixing and heat transfer between the jet and the co-flow and the spreading of the jet wee the subjects of further investigations.

## 4. Predictions with Different Chemical Kinetic Mechanisms

As previously mentioned, the EDC has been successfully applied to the numerical simulation of combustion for a long period of time, and it is implemented in most available commercial CFD codes. Various models have been validated with the Flame D setup with the detailed chemical kinetic mechanism of GRI-Mech 3.0 [15,32–39].

The expense of using a very detailed chemical kinetics mechanism, such as GRI-Mech 3.0, is, however, in general quite high and might be unacceptable for engineering purposes. One way of reducing this cost is by applying mechanism reduction methods that neglect the less important elementary reactions in the detailed mechanism during computation. A second approach is directly applying simplified reaction mechanisms, such as ARM 9 [49] and ARM 19 [49], or even a simple global reaction mechanism [32,56]. The objective of this section was to investigate the performance of the chemical kinetics mechanisms with different species and elemental reactions from GRI-Mech in comparison to the experimental results when simulating a methane–air non-premixed flame with the EDC at a reasonable computational cost. In order to compare the performance of the mechanisms, a series of calculations was conducted for methane–air combustion with RSM turbulence model, which was validated with the detailed mechanisms in the last section. Five relatively simple mechanisms and the detailed mechanism of GRI-Mech 3.0 were chosen for comparison with the experiments [40–42]. The details of the mechanisms used in the validation, including the corresponding cases, are given in Table 4. The results for the Sandia Flame D from the six mechanisms were measured with the EDC model.

**Table 4.** Chemical mechanisms used in the calculations.

| Mechanism | No. of Species | No. of Steps | NO Species | Case | Reference |
|-----------|----------------|--------------|------------|------|-----------|
| GRI3.0 | 53 | 325 | With NO | EDC-53 | [44] |
| GRI2.11 | 49 | 279 | With NO | EDC-49 | [46] |
| GRI1.2 | 32 | 177 | Without NO | EDC-32 | [45] |
| sp24 | 24 | 104 | Without NO | EDC-24 | [47] |
| sp21 | 21 | 84 | Without NO | EDC-21 | [47] |
| Skeletal | 16 | 41 | Without NO | EDC-16 | [51] |

A partially premixed combustion (PPC) model, which is one of the common combustion modes for many practical combustion systems, is based on the non-premixed combustion model and the premixed combustion model [37,74–76]. Since the PPC PDF model (referred to as the PDF model below) with chemical equilibrium for state relation and adiabatic energy treatment is widely used in industry-scale turbulent combustion simulations [77,78], the results with this model are also presented for comparison due to its simplicity and very fast simulation. One thing must be noted: the inlet pilot mixture composition could not be exactly defined in the PDF model, which might have resulted in some uncertainties. The only difference of the work in this section was the reaction mechanism used for each case.

The sensitivity to the reaction mechanism was investigated by applying EDC-53, EDC-49, EDC-32, EDC-24, EDC-21, EDC-16, or PDF with the RSM-based turbulence model and compared with the experimental data [40–42] to determine the candidate reaction mechanism to be used in the simulations of industry-scale natural gas combustion furnaces. The details of the cases for EDCs can be found in Table 4, and the settings were the same as those in the last section.

A comparison of the predicted flame temperatures is displayed in Figure 6. The predicted flame constructions by the chemical kinetics mechanisms were found to be quite similar. Compared to the EDC models, the result from the PDF model, shown in Figure 6g, showed little difference regarding flame shape (the high temperature area). A

low temperature was found at the pilot exit, and the flame shape was thinner compared to others. Since the used turbulence model was the same for all of them, the differences between them must have mainly resulted from the chemical reaction modelling.

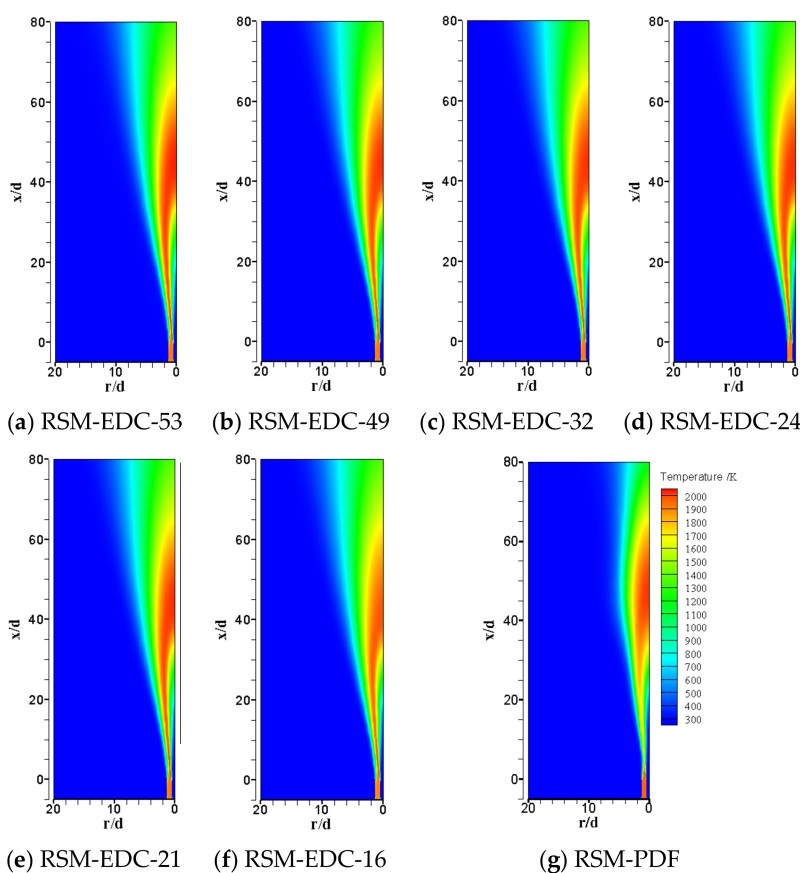

(**a**) RSM-EDC-53    (**b**) RSM-EDC-49    (**c**) RSM-EDC-32    (**d**) RSM-EDC-24

(**e**) RSM-EDC-21    (**f**) RSM-EDC-16      (**g**) RSM-PDF

**Figure 6.** Iso-contours of mean temperature for the Sandia Flame D with different mechanisms for methane–air combustion.

Deviations of predicted and measured values are shown in Table 5 for different combustion models with the RSM model, where the largest deviation happened to the minor species, i.e., CO, $H_2$, and OH. The main scalar data of flow from these seven cases are presented in Figures 7–9. The mean temperatures along the axis and along the radius @$x/d$ = 7.5, 30, or 45 are shown and compared with experiments in Figure 7 for the Sandia Flame D with different reaction treatments of the methane–air combustion simulation. Once again, since the used turbulence model was the same for all of them, the deviations between the predictions and the experiments shown in Figures 7–9 and Table 5 must have mainly resulted from the chemical reaction modelling. All the axial mean temperature peaks were slightly over-predicted, and the largest was about 138 K by EDC-53. The predicted axial mean temperature peaks and locations were ~2083 K@$x/d$~45 for EDC-53, ~2052 K@ $x/d$~45 for EDC-49, ~2054 K@$x/d$~45 for EDC-32, ~2057 K@$x/d$~45 for EDC-24, ~2058 K@$x/d$~45 for EDC-21, ~1986 K@$x/d$~41 for EDC-16, and ~2045 K@$x/d$~48 for PDF in Table 5. The peak temperature locations were properly predicted with five EDCs. On the other hand, the peak with EDC-16 was shifted towards the burner (Figure 7a), and the peak with PDF was shifted downwards from the burner a little (Figure 7a) compared to the experiments (~1945 K@$x/d$~45). These deviations and others, shown in Figures 4 and 5 and Table 5, are also found by other predictions [34,37]. For the radial mean temperature profiles shown in Figure 7b–d indicate that the results from the EDCs were largely over-predicted, and the results from PDF agreed better with the experiments. The results by the

PDF model, compared with those by the EDCs, showed apparent lower values of the mean temperature, especially the radial profiles.

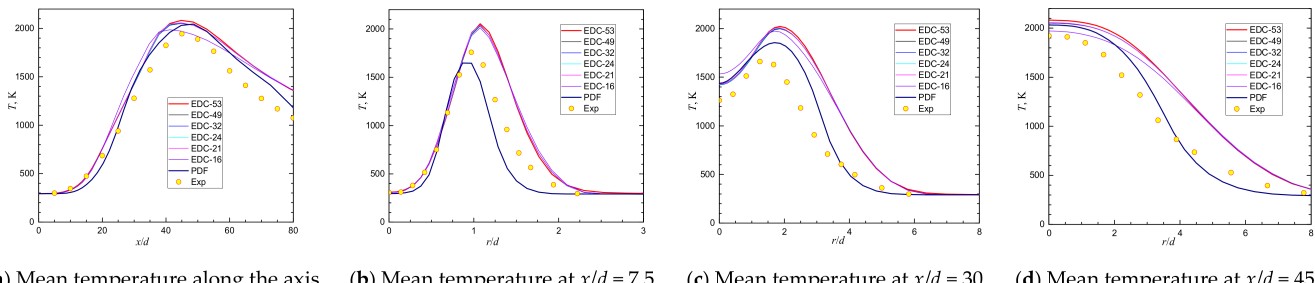

(**a**) Mean temperature along the axis    (**b**) Mean temperature at $x/d$ = 7.5    (**c**) Mean temperature at $x/d$ = 30    (**d**) Mean temperature at $x/d$ = 45

**Figure 7.** Comparison of predicted and measured mean temperature for the Sandia Flame D.

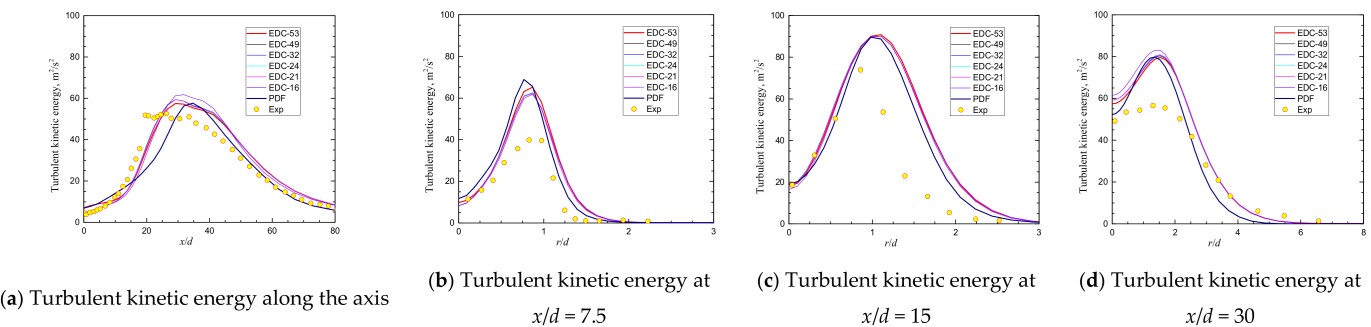

(**a**) Turbulent kinetic energy along the axis    (**b**) Turbulent kinetic energy at $x/d$ = 7.5    (**c**) Turbulent kinetic energy at $x/d$ = 15    (**d**) Turbulent kinetic energy at $x/d$ = 30

**Figure 8.** Comparison of predicted and measured turbulent kinetic energy for the Sandia Flame D.

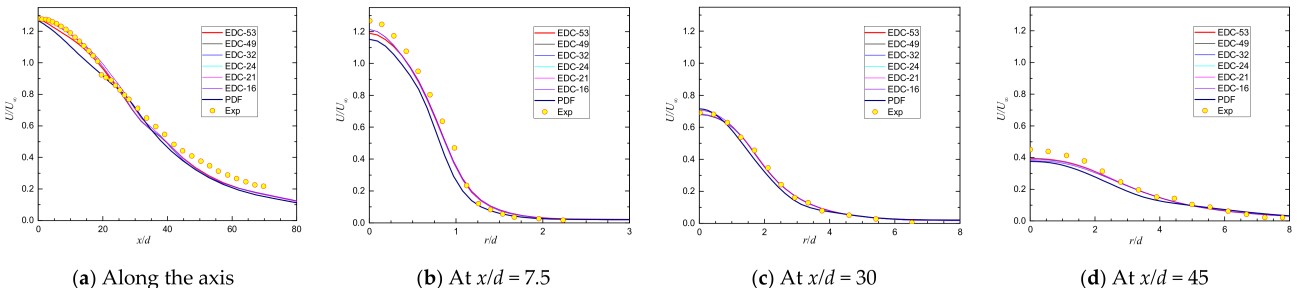

(**a**) Along the axis    (**b**) At $x/d$ = 7.5    (**c**) At $x/d$ = 30    (**d**) At $x/d$ = 45

**Figure 9.** Comparison of predicted and measured mean axial velocity for the Sandia Flame D.

Generally, a reasonable qualitative prediction of the flame was achieved for all model combinations, especially the axial temperature profiles. However, in contrast to those results from LES simulations [15,33–35,37,38], none of them came quantitatively close to the measured data. All the axial temperature profiles by the EDCs shown in Figure 7a agreed very well with the experiment data before the axial location of $\sim x/d$ = 20, and the others were a little over-predicted.

Turbulent kinetic energies along the axis and along the radius @$x/d$ = 7.5, 15, or 30 are shown and compared with the experiments in Figure 8 for the Sandia Flame D with different reaction treatments of the methane–air combustion simulation. All the predictions of the turbulent kinetic energies from the seven cases were a little over-predicted, especially the peaks. Once again, all the predictions by the EDCs agreed very well with each other. The axial profile of the turbulent kinetic energy by the PDF model was found to agree well with the experiments downstream the temperature peak at $x/d\sim$45. Generally, a better agreement with the experiments was reached by the PDF model among the seven cases.

**Table 5.** Deviations of predicted and measured values for different combustion models with the RSM turbulence model (deviation = (prediction − measurement)/measurement × 100%).

| Scalar Peak and Its Location | Exp | RSM-EDC-53 | Deviation | RSM-EDC-49 | Deviation | RSM-EDC-32 | Deviation | RSM-EDC-24 | Deviation | RSM-EDC-21 | Deviation | RSM-EDC-16 | Deviation | PDF | Deviation |
|---|---|---|---|---|---|---|---|---|---|---|---|---|---|---|---|
| Max Temperature@axis | 1945 | 2083 | 7.08% | 2052 | 5.52% | 2054 | 5.61% | 2057 | 5.76% | 2058 | 5.81% | 1986 | 2.12% | 2045 | 5.13% |
| Location@axis, $x/d=$ | 45.00 | 44.75 | | 44.75 | | 44.75 | | 44.75 | | 44.75 | | 41.16 | | 48.62 | |
| Max Turbulent energy@axis | 52.57 | 57.50 | 9.38% | 59.42 | 13.03% | 59.39 | 12.99% | 59.33 | 12.86% | 59.37 | 12.94% | 61.78 | 17.53% | 55.23 | 5.07% |
| Location@axis, $x/d=$ | 26.28 | 29.20 | | 29.20 | | 29.20 | | 29.20 | | 29.20 | | 31.86 | | 34.74 | |
| Min $YO_2$@axis | 0.019 | 0.006 | −69.93% | 0.006 | −66.91% | 0.006 | −67.21% | 0.006 | −68.54% | 0.006 | −70.40% | 0.005 | −75.24% | 0.007 | −64.81% |
| Location@axis, $x/d=$ | 45.00 | 41.16 | | 37.83 | | 37.83 | | 37.83 | | 37.83 | | 37.83 | | 37.83 | |
| Max $YH_2O$@axis | 0.116 | 0.123 | 6.02% | 0.123 | 5.74% | 0.123 | 6.07% | 0.124 | 6.92% | 0.125 | 7.40% | 0.122 | 4.82% | 0.116 | 0.23% |
| Location@axis, $x/d=$ | 45.00 | 37.83 | | 37.83 | | 37.83 | | 37.83 | | 37.83 | | 37.83 | | 44.75 | |
| Max $YCO_2$@axis | 0.108 | 0.118 | 8.95% | 0.115 | 6.78% | 0.115 | 6.81% | 0.115 | 6.83% | 0.115 | 6.52% | 0.108 | 0.28% | 0.115 | 6.29% |
| Location@axis, $x/d=$ | 50.00 | 48.62 | | 48.62 | | 48.62 | | 48.62 | | 48.62 | | 44.75 | | 52.79 | |
| Max $YCO$@axis | 0.0453 | 0.0565 | 24.62% | 0.0574 | 26.80% | 0.0572 | 26.36% | 0.0561 | 23.81% | 0.0545 | 20.33% | 0.0678 | 49.76% | 0.1094 | 141.57% |
| Location@axis, $x/d=$ | 40.00 | 37.83 | | 37.83 | | 37.83 | | 37.83 | | 37.83 | | 34.74 | | 31.86 | |
| Max $YH_2$@axis | 0.0029 | 0.0034 | 19.07% | 0.0034 | 19.44% | 0.0034 | 17.64% | 0.0032 | 12.78% | 0.0029 | 0.92% | 0.0037 | 28.80% | 0.0110 | 281.44% |
| Location@axis, $x/d=$ | 40.00 | 34.74 | | 34.74 | | 34.74 | | 34.74 | | 34.74 | | 34.74 | | 29.20 | |
| Max $YOH$@axis | 0.0015 | 0.0040 | 171.18% | 0.0043 | 192.88% | 0.0043 | 191.90% | 0.0043 | 190.11% | 0.0043 | 191.66% | 0.0052 | 251.23% | 0.0009 | −36.82% |
| Location@axis, $x/d=$ | 50.00 | 44.75 | | 44.75 | | 44.75 | | 44.75 | | 44.75 | | 44.75 | | 48.62 | |

Mean axial velocity profiles along the axis and along the radius @$x/d$ = 7.5, 30, or 45 are shown and compared with experiments in Figure 9 for the Sandia Flame D with different reaction mechanisms. Reasonable qualitative predictions were reached for the mean axial velocity profiles from the seven cases, while the predictions by the PDF model were a little under-predicted. Very small differences of mean axial velocity profiles, shown in Figure 9, were found from the predictions by the EDCs. The largest deviations were found downstream of the jet, as shown in Figure 9a,b close to the axis. This means the mixing between the jet and the pilot flows was over-predicted compared with the experiments, leading to the jet diffusing faster in this large temperature difference case.

Composition profiles of major species from these seven cases are presented in Figures 10–14 for the mass fractions of $CH_4$, $O_2$, $H_2O$, $CO_2$, and $N_2$, respectively. The mean $CH_4$ composition profiles along the axis and along the radius @ $x/d$ = 7.5 and 30 are shown and compared with experiments in Figure 10 for the Sandia Flame D with different reaction mechanisms. The mean $O_2$ composition profiles along the axis and along the radius @ $x/d$ = 7.5, 30, and 45 are shown and compared with experiments in Figure 11. The mean $H_2O$ composition profiles along the axis and along the radius @ $x/d$ = 7.5, 30, and 45 are shown and compared with experiments in Figure 12. The mean $CO_2$ composition profiles along the axis and along the radius @ $x/d$ = 7.5, 30, and 45 are shown and compared with experiments in Figure 13. The mean $N_2$ composition profiles along the axis and along the radius @ $x/d$ = 7.5, 30, and 45 are shown and compared with experiments in Figure 14. Deviations of predicted and measured values are shown in Table 5 for different combustion models with the RSM model the major species, where the largest deviation happened to the peak values of the $O_2$ mass fraction shown in Figure 11a. The minimum $O_2$ mass fraction profiles shown in Figure 11a were acceptable with the exception of a little shifting towards the burner. Again, all the predictions by the EDCs agreed very well with each other. Apparent differences of the profiles were found from the EDC-16 and PDF predictions. Most of the predictions of the mean composition profiles from the seven cases were a little over/under-predicted, especially the peaks. The best agreement with the experiments among the seven cases for the profiles of major species, with the exception of $CH_4$, was reached by the PDF model.

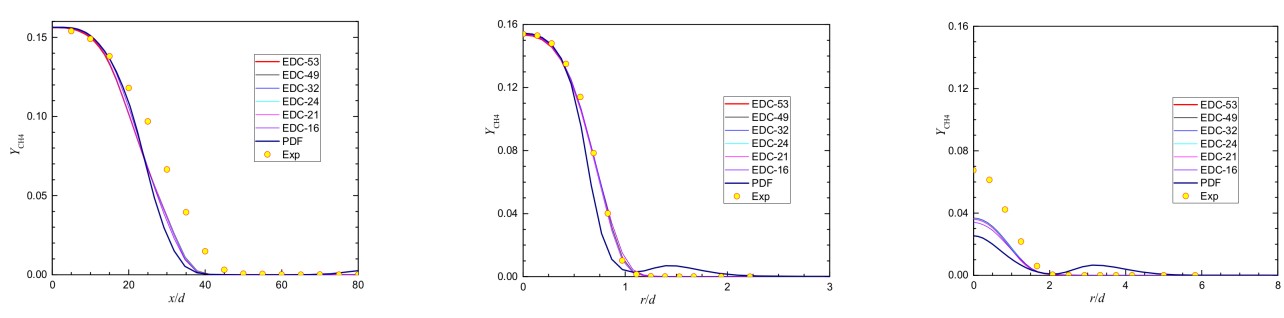

(**a**) Mean $CH_4$ composition along the axis    (**b**) Mean $CH_4$ composition at $x/d$ = 7.5    (**c**) Mean $CH_4$ composition at $x/d$ = 30

**Figure 10.** Comparison of predicted and measured mean $CH_4$ mass fraction profiles for the Sandia Flame D.

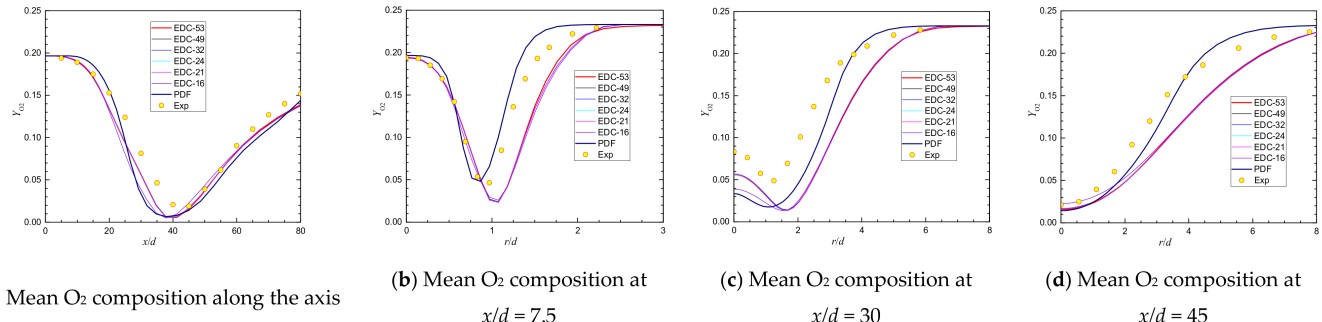

(**a**) Mean $O_2$ composition along the axis    (**b**) Mean $O_2$ composition at $x/d$ = 7.5    (**c**) Mean $O_2$ composition at $x/d$ = 30    (**d**) Mean $O_2$ composition at $x/d$ = 45

**Figure 11.** Comparison of predicted and measured mean $O_2$ mass fraction profiles for the Sandia Flame D.

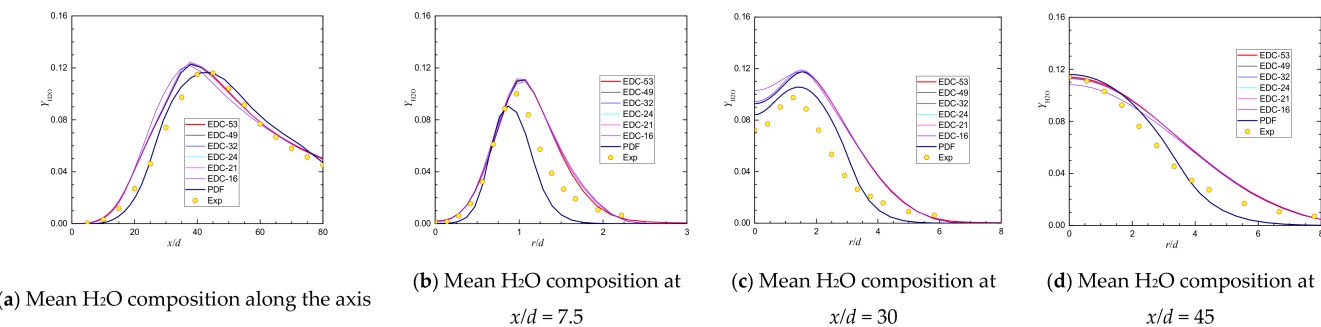

(**a**) Mean H₂O composition along the axis

(**b**) Mean H₂O composition at $x/d = 7.5$

(**c**) Mean H₂O composition at $x/d = 30$

(**d**) Mean H₂O composition at $x/d = 45$

**Figure 12.** Comparison of predicted and measured mean $H_2O$ mass fraction profiles for the Sandia Flame D.

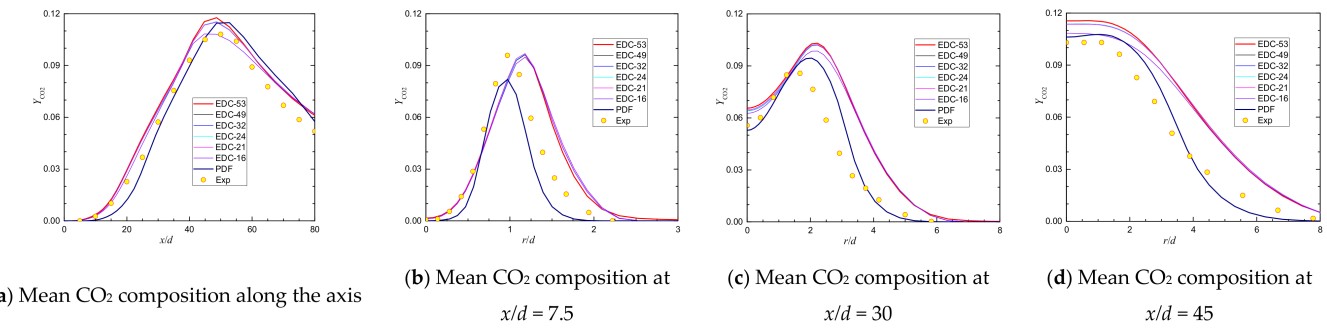

(**a**) Mean CO₂ composition along the axis

(**b**) Mean CO₂ composition at $x/d = 7.5$

(**c**) Mean CO₂ composition at $x/d = 30$

(**d**) Mean CO₂ composition at $x/d = 45$

**Figure 13.** Comparison of predicted and measured mean $CO_2$ mass fraction profiles for the Sandia Flame D.

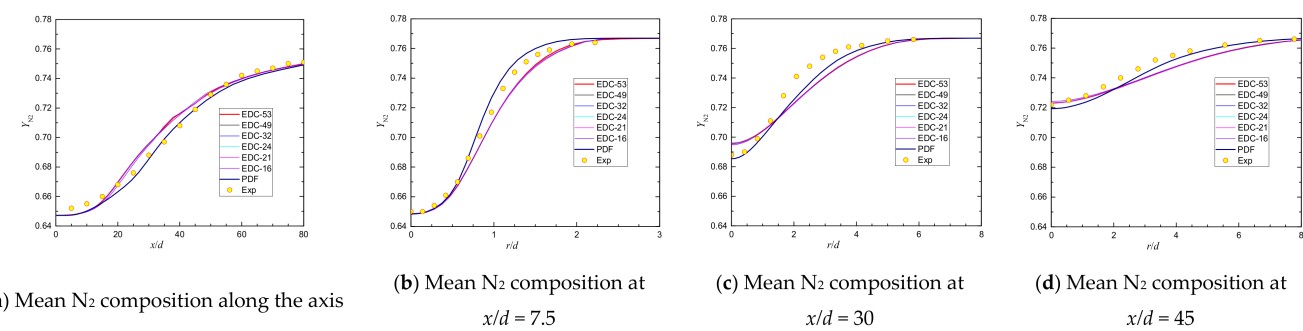

(**a**) Mean N₂ composition along the axis

(**b**) Mean N₂ composition at $x/d = 7.5$

(**c**) Mean N₂ composition at $x/d = 30$

(**d**) Mean N₂ composition at $x/d = 45$

**Figure 14.** Comparison of predicted and measured mean $N_2$ mass fraction profiles for the Sandia Flame D.

The predicted composition profiles of the reactant mass fractions of $CH_4$ and $O_2$, shown in Figures 10 and 11, were lower than the experiments and those of the product mass fractions of $H_2O$ and $CO_2$ were larger (as shown in Figures 12 and 13), indicating that combustion progressed faster than that in the experiment. The $CH_4$ composition profiles from the PDF model, shown in Figure 10, suggested a small amount of $CH_4$ re-formation close to the axis. Due to the over-predicted mixing and diffusion of the jets, the data close to the axis were smaller than those of the experiments shown in Figure 10c. The formation of $H_2O$ by the EDCs was earlier than in the experiments, as shown in Figure 12a with quantitative agreement and a little over-predicted peak. Almost all the radial predictions of $H_2O$ were larger than the experiments at $x/d = 30$, as shown in Figure 12c, meaning that the combustion was intensive at this place with a very similar shape of the temperature, as shown in Figure 7c. Other major composition profiles had similar distributions of the composition of $H_2O$, as shown in Figure 11 for $O_2$ and Figure 13 for $CO_2$. The inert $N_2$ composition profiles, shown in Figure 14, agreed quantitatively well with the experiments. Once again, all the $N_2$ predictions by the EDCs agreed very well with each other, and the

best results were from the PDF model. The differences between simulations and measured data were not significant for the temperature, the main species (such as $H_2O$, $O_2$, $N_2$, and $CO_2$), and the mean velocities. The behavior of these species had the same trend as that for the temperature. The profiles of the values for composition along the axis and the radius replicated the behavior of the temperature.

The composition profiles of minor species of the mass fractions of $H_2$, CO, and OH from these seven cases are presented in Figures 15–17, respectively, and the deviations of predicted and measured values are shown in Table 5 for different combustion models with the RSM model. The mean $H_2$ composition profiles along the axis and along the radius @ $x/d$ = 7.5, 30, and 45 are shown and compared with experiments in Figure 15 for the Sandia Flame D with different reaction mechanisms. The mean CO composition profiles along the axis and along the radius @ $x/d$ = 7.5, 30, and 45 are shown and compared with experiments in Figure 16. The mean OH composition profiles along the axis and along the radius @ $x/d$ = 7.5, 30, and 45 are shown and compared with experiments in Figure 17. Small differences were found from the predictions by the EDCs for the composition profiles of minor species. Apparent differences of the profiles, shown in Figures 15–17 and Table 5, were found from the PDF predictions. Most of the predictions of the mean composition profiles from the seven cases were a little over/under-predicted, especially the peaks of the PDF model. Regarding the profiles of the minor species, such as OH, it should also be noted that the choice of the chemical reaction modelling also affected the prediction of the formation and peak values. The predictions of the minor species, $H_2$, CO, and OH, are shown and compared in Figures 15–17 and Table 5. The prediction of OH in any combustion simulation is particularly challenging due to the strong nonlinearity of the species' evolution [79]. Compared with the respected measured data, the level of agreement displayed by OH was not very good, with largely over-predicted peaks with the EDC models and under-predicted peaks with the PDF model, as shown in Figure 17a–d.

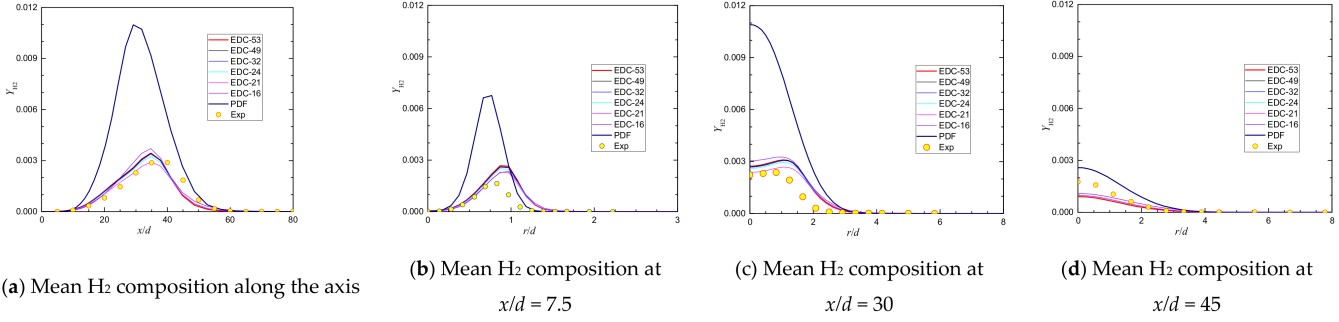

(**a**) Mean $H_2$ composition along the axis

(**b**) Mean $H_2$ composition at $x/d$ = 7.5

(**c**) Mean $H_2$ composition at $x/d$ = 30

(**d**) Mean $H_2$ composition at $x/d$ = 45

**Figure 15.** Comparison of predicted and measured mean $H_2$ mass fraction profiles for the Sandia Flame D.

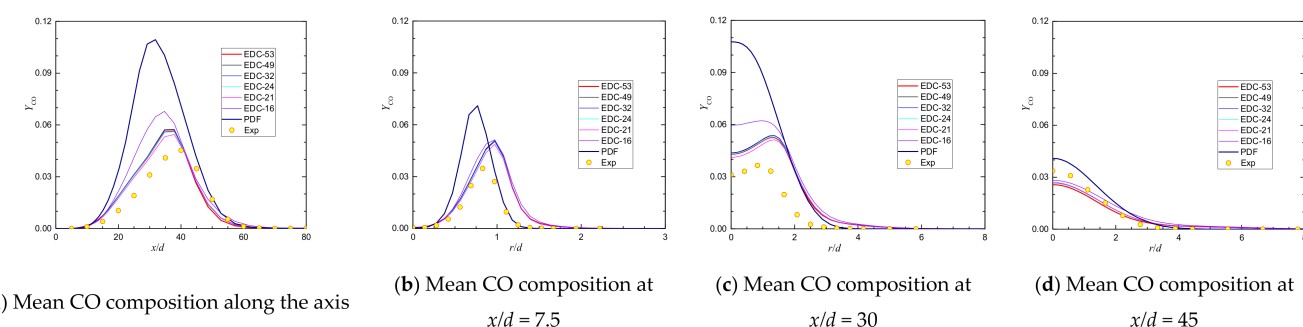

(**a**) Mean CO composition along the axis

(**b**) Mean CO composition at $x/d$ = 7.5

(**c**) Mean CO composition at $x/d$ = 30

(**d**) Mean CO composition at $x/d$ = 45

**Figure 16.** Comparison of predicted and measured mean CO mass fraction profiles for the Sandia Flame D.

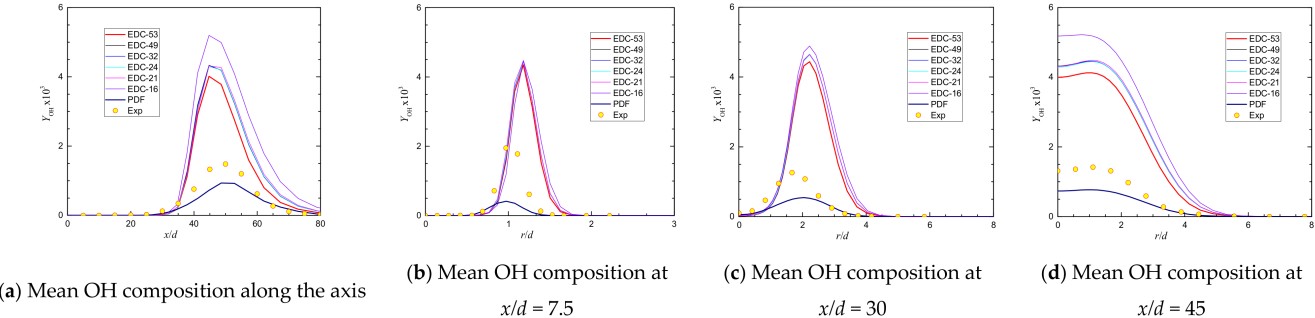

(**a**) Mean OH composition along the axis

(**b**) Mean OH composition at $x/d = 7.5$

(**c**) Mean OH composition at $x/d = 30$

(**d**) Mean OH composition at $x/d = 45$

**Figure 17.** Comparison of predicted and measured mean OH mass fraction profiles for the Sandia Flame D.

One of the possible reasons for the discrepancies of major and minor species, shown in Figures 10–17 and Tables 4 and 5, may have been the inaccurate treatments of heat radiation, which had a significant effect on the predicted mean compositions. Some of the predicted maximum/minimum mass fractions, shown in Figures 10–17 and Tables 4 and 5, were significantly over/under-predicted compared with the respected measured data, which was probably due to the temperature overestimation by the approximate radiation heat transfer treatment when using the DO model along with the WSGGM for absorption coefficients.

For the objective of this section, we chose the prediction results by EDC-53 as the comparison base, and the other simulation results were compared with such a base. Generally, the predictions agreed well with those by EDC-53 for the mean temperature, the turbulent kinetic energy, the mean axial velocity, and the compositions for the major species, with exceptions of those by the PDF model. There were some disagreements between the predictions for the axial and the radial distributions for the temperature and the compositions for the Sandia Flame D. Among the EDCs, the results from the smaller number of species by EDC-21 and EDC-24 agreed very well with the results by EDC-53, the detailed chemical kinetic mechanism of GRI-Mech 3.0 for methane–air combustion.

The results, shown in Figures 6–14 and Table 5, indicated that the EDCs could give relatively good predictions for the turbulent piloted methane–air diffusion flame of the Sandia Flame D. For the simulations of industry-scale natural gas combustion furnaces, relative simpler chemical kinetic mechanism and lesser computational costs, such as shown by EDC-21 or EDC-24, could be used for combustion simulations at affordable expenses. Since the result of EDC-21 was quantitatively closest to that of EDC-53, as clearly shown in Figures 6–14, EDC-21 could be used for the simulation of the turbulent piloted methane–air diffusion flame, the Sandia Flame D. With more validations, EDC-21 might be used for simulations in engineering applications. On the other hand, most of the results by the PDF model with lesser simulation expenses than EDC-21, as shown in Figures 6–14, agreed better with the experiments. However, NO composition was not included in the EDCs with exceptions of EDC-53 and EDC-49. If the emission of NO has to be predicted during the turbulent methane–air combustion, an $NO_x$ sub-model (implemented in most commercial CFD codes) could be used after validation. Therefore, the PDF model is suggested for simulations of the engineering applications of methane–air combustion based on the results shown in Figures 6–14 and the deviations shown in Table 5.

The disagreements between the measured and the computed scalar values might have mainly been due to the reasons mentioned in this work and partly due to the treatments of reactions, which will the subjects of further investigations.

## 5. Conclusions

Comprehensive numerical investigations were carried out in order to set-up a CFD model for the more accurate prediction of turbulent reacting flow for the Sandia flame D than by the commonly used simple turbulence–chemistry interaction models coupled with global reaction kinetics in the simulations of industry-scale natural gas combustion furnaces. The EDC and the PDF models were applied to RAS of the turbulent methane–air

combustion for the Sandia Flame D. The validated turbulence models were the SKE, the SKE-R with the standard constant changed, the RKE model, the SST $k$-$\omega$ model, and the RSM. The predictions were compared to experimental data from the literature.

The $k$-$\varepsilon$ models and the RSM model are widely used in engineering simulations. For the validation of turbulence models, the predicted profiles by RSM with the EDC-based detailed chemical kinetic mechanism of GRI-Mech 3.0 more closely agreed with the experiments than those by SKEs and SST. Very small differences were found from the results of the turbulence models of RKE and SST with the EDC-based detailed chemical kinetic mechanism. The predicted scalar data by RSM were very close to the experiments for the turbulence simulation of the methane–air combustion.

Different treatments of methane–air combustion were compared for the Sandia Flame D with the RSM model. The results from seven cases showed reasonably good agreements with the experiments. For the simulations of industry-scale natural gas/syngas combustion furnaces, therefore, partially premixed combustion with the state relation chemistry of chemical equilibrium, the mixture fraction PDF model, and the RSM are recommended for their relatively very low simulation expenses, acceptable accuracy predictions, and quantitatively good agreement with the experiments from this work.

Deviations were also found between the predictions and the experiments for the turbulent methane–air combustion of the Sandia flame D. There were some sources for to the deviations that over-predicted the mixing and the diffusion, including simplifications of the boundary settings, the turbulence model, the turbulence–reaction interaction, and the radiation heat transfer. For $NO_x$ modelling in industry-scale methane–air combustion furnaces, a validated $NO_x$ sub-model, implemented in most commercial CFD codes, could be used in engineering methane–air combustion.

**Author Contributions:** Conceptualization, Y.Y. and C.W.; methodology, D.H.; validation, D.H. and Y.K.; formal analysis, D.H.; investigation, D.H. and Y.K.; writing—original draft preparation, D.H.; writing—review and editing, C.W., Y.Y., and Y.K.; supervision, Y.Y.; project administration, Y.Y. All authors have read and agreed to the published version of the manuscript.

**Funding:** This research was funded by the National Natural Science Foundation of China, 51576014.

**Institutional Review Board Statement:** Not applicable.

**Informed Consent Statement:** Not applicable.

**Data Availability Statement:** Not applicable.

**Conflicts of Interest:** The authors declare no conflict of interest.

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
