# Peer review of "Model Comparisons of Flow and Chemical Kinetic Mechanisms for Methane–Air Combustion for Engineering Applications"

_applsci, doi:10.3390/app11094107_

Round 1
Reviewer 1 Report
Dear Authors,
Many thanks for your submitted work. It was really interesting when you summarized models with different turbulent models and ..... I enjoyed your work, however, I have some suggestions for you to make your work more step forward. Please, find below comments on your work:
- Since your title is talking about engineering application, I would rather to see in the first or second paragraph that you talk about several engineering applications and properly cite them.
- After the first or second paragraph, You could also discuss and classify your introduction to talk about some subtopics that used Methane Combustion like: Swirling and Non-Swirling, Internal or external combustion, laminar and turbulent, furnaces, Turbulent models, Kinetic models, Turbulent and kinetic models together.
- In my opinion your references are very old and not updated! there are plenty works similar to your work which talked about the Methane combustion! Please used last 5 years papers! Like: " Erfan Khodabandeh, Hesam Moghadasi, Mohsen Saffari Pour, Mikael Ersson, Pär G. Jönsson, Marc A. Rosen, Alireza Rahbari,
CFD study of non-premixed swirling burners: Effect of turbulence models,
Chinese Journal of Chemical Engineering, Volume 28, Issue 4, 2020,Pages 1029-1038, https://doi.org/10.1016/j.cjche.2020.02.016. - I couldnt find your novelty! Because the turbulent models and all related to EDC models or GRI have been properly discussed in many references. You need to be more specific and classified as I mentioned before.
- Your conclusions are also very general! They have been properly discussed in many papers!
- Please avoid using in general in scientific paper, more specifically in Abstract. Please be more specific!
- Maybe would be better to present some contours for pressure and some fractionations as well to show your accuracy and also the differences between models visually!
Author Response
Dear Editor,
We are very pleased to been given the opportunity to revise our manuscript. The authors would like to express their appreciation to all the reviewers. Thanks for the reviewers’ useful comments and suggestions. The reviewers provide us very valuable suggestions and we have modified the manuscript accordingly. The detailed responses are listed below point by point.

Reviewer 2 Report
The paper concerns the modeling of the Sandia Flame D. Different turbulence models are compared and evaluated using the detailed methane-air reaction mechanism GRI 3.0.
The computational mesh is very coarse and the study of grid independence must be carried out over a wider range of grids with a considerably higher grid resolution. The present study is not sufficient to prove grid independence. The statement that a much finer grid might be unacceptable for engineering applications does not prove the correctness of the numerical solution for coarse grids.
The pilot mixture composition must be provided in Tab. 1 or somewhere else. Where do the estimated inlet profiles that are referred to in Tab. 1 come from? How were they estimated?
Figs. 3 and 6: provide unit in the legend of the temperature.
An adjustment of parameter constants in the SKE model to SKE-R is not very reasonable since it violates the thought of predictive simulation capacity.
What does ‘production limiter’ in Tab. 2 refer to?
The paper gives not much new information. The most advanced combustion model performs best which is not much surprising.
Author Response

(The authors gave the same response as above.)

Round 2
Reviewer 1 Report
The revised version is much more better than the original one, when they try to describe their work with reasonable scientific background.